# A Method for Autonomous Generation of High-Precision Time Scales for Navigation Constellations

**DOI:** 10.3390/s23031703

**Published:** 2023-02-03

**Authors:** Shitao Yang, Xiao Yi, Richang Dong, Qianyi Ren, Xupeng Li, Tao Shuai, Jun Zhang, Wenbin Gong

**Affiliations:** 1University of Chinese Academy of Sciences, Beijing 100049, China; 2Innovation Academy for Microsatellites of CAS, Shanghai 200120, China; 3Shanghai Engineering Center for Microsatellites, Shanghai 200120, China; 4The 29th Research Institute of China Electronics Technology Group Corporation, Chengdu 610036, China; 5Shanghai Astronomical Observatory of Chinese Academy of Sciences, Shanghai 200030, China

**Keywords:** time scale algorithm, time-frequency steering algorithm, digital phase locked loop, frequency stability, genetic algorithm, autonomous navigation

## Abstract

The time maintenance accuracy of the navigation constellation determines the user positioning and timing performance. Especially in autonomous operation scenarios, the performance of navigation constellation maintenance time directly affects the duration of constellation autonomous navigation. Among them, the frequency stability of the atomic clock onboard the navigation satellite is a key factor. In order to further improve the stability of the navigation constellation time-frequency system, combined with the development of high-precision inter-satellite link measurement technology, the idea of constructing constellation-level synthetic atomic time has gradually become the development trend of major GNSS systems. This paper gives a navigation constellation time scale generation framework, and designs an improved Kalman plus weights (KPW) time scale algorithm and time-frequency steer algorithm that integrates genetic algorithms. Finally, a 30-day autonomous timekeeping simulation was carried out using the GPS precision clock data provided by CODE, when the sampling interval is 300 s, the Allan deviation of the output time scale is 5.73 × 10^−14^, a 71% improvement compared with the traditional KPW time scale algorithm; when the sampling interval is 1 day, the Allan deviation is 9.17 × 10^−15^; when the sampling interval is 1 × 10^6^ s, the Allan deviation is 8.87 × 10^−16^, a 94% improvement compared with the traditional KPW time scale algorithm. The constellation-level high-precision time scale generation technology proposed in this paper can significantly improve the stability performance of navigation constellation autonomous timekeeping.

## 1. Introduction

The basic principle of the global navigation satellite system (GNSS) is timing and ranging. In order to ensure that GNSS provides users with high-precision navigation positioning and timing services, it is necessary to establish and maintain a high-precision time reference [1]. At present, the time reference of mainstream GNSS is established and maintained by the atomic clock group of the ground master station, such as GPS System Time (GPST) and Beidou System Time (BDT). The satellite-borne atomic clocks on the navigation satellites obtain the satellite-ground clock difference through the satellite-ground two-way time comparison technology, and achieves the satellite-ground time synchronization [2]. With the development of the autonomous operation technology of the navigation constellation, in order to maintain the continuity of the time-frequency signal of the GNSS and reduce the dependence on the ground station, the navigation constellation needs to independently establish and maintain a high-precision time scale as the time-frequency reference of each single satellite [3]. The time difference of an atomic clock can be composed of a deterministic trend term and a random term error: ($x\left(t\right)=x_{0}+y_{0}t+\frac{Dt^{2}}{2}+\varepsilon \left(t\right)$), where the deterministic trend term includes the initial time error $x_{0}$, the initial frequency deviation is $y_{0}$, and the linear frequency drift D is caused by aging and temperature changes. ε(t) is the random frequency deviation of different types of noise such as stationary noise and non-stationary noise, mainly including: WPM, FPM, WFM, FFM and RWFM noise. Since the atomic clock is composed of various electronic components, the five noise levels of each clock are not exactly the same, and the frequency stability and frequency accuracy are also different. Therefore, the random noise of the output time scale can be reduced by the time scale algorithm, and the frequency stability can be improved.

With the development of inter-satellite link technology, the clock difference accuracy obtained by two-way measurement of the inter-satellite link is less than 0.3 ns [2,4]. Therefore, the satellite clocks can be formed into a large clock group through the inter-satellite link, and the time scale algorithm can be used to obtain a more stable time scale [5]. Currently, commonly used time scale algorithms include weighted average algorithm (ALGOS algorithm and AT1 algorithm) [6], Kalman filter algorithm [7], Kalman plus weights (KPW) algorithm [8,9], wavelet decomposition algorithm, etc. The core idea of the ALGOS algorithm and the AT1 algorithm is to assign a weight to each atomic clock, thereby increasing the stability of the time scale. However, without eliminating or suppressing the noise of the atomic clock, it cannot further improve the accuracy and reliability of the atomic time scale. For the Kalman filter time scale algorithm, it can model a variety of noises and suppress these noises at the same time. Its obtained time scale is significantly better than the ALGOS algorithm that can only mainly suppress one kind of noise, but its incompleteness will cause the error of the estimation result to increase with time, and the accuracy of the time scale will decrease. The KPW algorithm proposed by Greenhall assumes that only white frequency noise (WFM) remains in the clock data after Kalman frequency correction, and adjusts the weight according to the strength of WFM, thus it can take into account both short-term and long-term stability of the time scale. Since different spaceborne atomic clocks have different time-frequency characteristics, the scheme in this paper is to divide multiple rubidium atomic clocks and cesium atomic clocks into two clock groups according to type, and use the genetic algorithm [10] to improve the KPW time scale algorithm. The two clock groups use this time scale algorithm to generate two timescales. Then, the improved time-frequency steering algorithm is used to obtain a comprehensive time scale that takes into account the time-frequency characteristics of the two types of atomic clocks.

In the steering experiment, the deterministic trend items, $x_{0}$, $y_{0}$, D, are removed first. The random noise of the atomic clock (including: five kinds of noise) is controlled, and the purpose is to combine the steering reference in the frequency domain with the excellent near-end noise level and the controlled frequency. Target good far-end noise levels. In terms of time domain, it is mainly reflected in the advantages of both frequency stability and long-term stability. Short-term stability and long-term stability correspond to the five types of noise corresponding to different sampling intervals. The time-frequency steering algorithm includes open-loop control and closed-loop control. The general open-loop control adopts the clock difference prediction method of the Kalman filter, and corrects the model prediction parameters to obtain the adjustment value. It does not depend on the measurement, but it can only improve the time synchronization accuracy. In the closed-loop control algorithm, the early zero-return phase modulation method will cause phase jumps, resulting in discontinuous time error data and deteriorating signal stability. The early ping-pong frequency modulation method based on two-state control used a specific frequency offset to make the time error signal oscillate between two thresholds to achieve phase continuity, but the result was not smooth [11]. The classic position-type PID closed-loop driving algorithm has a simple control structure, but the three parameters (proportional, integral and differential) that affect the degree of steering adjustment need to be adjusted manually and rely more on empirical values [12,13,14,15]. The traditional optimal quadratic Gaussian control algorithm (LQG) adjusts WQ and WR parameters through numerical simulation, and iteratively calculates the optimal time-frequency steer value, which has higher reliability [16,17], but its parameter selection process is more complicated. The Kalman filter combined with the delay structure can be equivalent to the closed-loop control method of the digital phase-locked loop (DPLL), which expands the application mode of the traditional DPLL [18,19]. The authors of references [20,21] selected the observable Kalman filter system, and verified the stability of the equivalent DPLL control system with a delay structure. In references [21,22], the standard second-order type 2 DPLL and the third-order type 3 DPLL are equivalently constructed. Because the adjustment parameters in references [21,23] need to be adjusted many times, there are precision limitations and self-adaptive approximate selection. Therefore, there is still room for optimization and improvement in the parameter selection of the algorithm. In this paper, the genetic optimization algorithm is used to realize the adaptive adjustment of parameters, and to promote the system to achieve the desired optimal steering goal with higher parameter selection accuracy and reliability.

In the second section, this paper will introduce the overall architecture of the navigation constellation high-precision time scale generation scheme. In the third section, the existing time scale algorithm is improved to obtain a time scale with higher stability than a single clock. In the fourth section, an improved time-frequency steering algorithm is introduced to generate a comprehensive time scale that takes into account both short-term and long-term stability. In the fifth section, the simulation results of the analytical scheme are summarized.

## 2. Time Scale Self-Generated Overall Architecture

The time scale generation scheme proposed in this paper uses each satellite-borne atomic clock in the navigation constellation as the time-frequency source, and each satellite-borne atomic clock forms two clock groups according to the type of clock through the inter-satellite link: the rubidium atomic clock group and the cesium atomic clock group. The relative clock difference can be calculated by inter-satellite link two-way ranging between satellites. Before the system enters the autonomous operation state, use the historical clock difference data of each atomic clock for five days to calculate the frequency stability, and select the atomic clock with the smallest Allan variance when the sampling interval is 10,000 s as the reference clock. Using the improved time scale algorithm, the time scale TA(Rb) of the rubidium clock group and the time scale TA(Cs) of the cesium clock group are respectively obtained. The frequency stability of two time scales is better than that of the single clock in the respective clock group.

In the part of time-frequency steer, select TA(Cs) as the control reference, TA(Rb) as the frequency standard to be steered, measure the relative clock difference between the control reference TA(Cs) and the TA(Rb) to be steered, and construct a third-order DPLLcontrol system equivalent to a Kalman filter combined with delay structure. On this basis, combined with the genetic algorithm to design the fitness function to control the adaptive iterative optimization of the system parameters, use the steer reference to obtain the optimal time-frequency steer value, and finally steer TA(Rb) efficiently to obtain the desired optimal comprehensive time scale. The structure diagram is shown in Figure 1.

## 3. Time Scale Algorithm Principle

The principle of the time scale algorithm is to firstly obtain the observed clock difference of each atomic clock in an atomic clock group relative to the reference clock. The observed clock difference is calculated to obtain a virtual clock difference sequence. The virtual clock difference sequence can be regarded as the time on paper. The stability of the time on the paper is better than any clock in the clock group. This section first introduces the KPW time scale algorithm, and then uses the genetic algorithm for optimization to obtain our desired time scale or paper time.

### 3.1. KPW Time Scale Algorithm

The paper time calculated by the time scale algorithm can be regarded as a virtual atomic clock, so it can be assumed that the clock difference between it and the ideal time scale is $x_{e}(t)$ [5], and the basic time scale equation is:(1)$$x_{e}(t)=\sum_{i=1}^{n}\omega_{i}(t)(x_{i}(t)-{\overset{}{\hat{x}}}_{i}(t))$$
where $\omega_{i}(t)$ represents the weight of each atomic clock, $x_{i}(t)$ represents the clock difference of each atomic clock, ${\overset{}{\hat{x}}}_{i}(t)$ represents the forecast value of each atomic clock difference; their deterministic trend terms have the same sign. Select an atomic clock with the best stability as a reference clock in the clock group, and its clock difference is $x_{j}(t)$, and the clock difference between it and the paper time is $x_{ej}(t)$:(2)$$x_{ej}(t)=x_{e}(t)-x_{j}(t)$$Substitute Equation (2) into Equation (1) to get:(3)$$x_{ej}(t)=\sum_{i=1}^{n}\omega_{i}(t)(x_{ij}(t)-{\overset{}{\hat{x}}}_{ie}(t))$$
where $x_{ij}(t)$ is the observed clock difference between each atomic clock and the reference clock, and ${\overset{}{\hat{x}}}_{ie}(t)$ is the clock difference prediction value of each atomic clock in the clock group relative to the virtual atomic frequency standard. According to the atomic clock difference model, ${\overset{}{\hat{x}}}_{ie}(t)$ can be expressed as
(4)$${\overset{}{\hat{x}}}_{ie}(t)={\overset{}{x}}_{ie}(t-\Delta t)+\tau \overset{}{{\hat{y}}_{\mathrm{ie}}(t-\Delta t)}+1/2D{\left(t-\Delta t\right)}^{2}$$
where Δt represents the calculation interval and D represents the frequency drift of the atomic clock. Combining Equations (3) and (4), the KPW algorithm equation [8] can be obtained as:(5)$$x_{ej}(t)=\sum_{i=1}^{n}\omega_{i}(t)(x_{\mathrm{ij}}(t)-({\overset{}{x}}_{ie}(t-\Delta t)+\Delta t\overset{}{{\hat{y}}_{\mathrm{ie}}(t-\Delta t)+1/2D{\left(t-\Delta t\right)}^{2})}$$It can be seen from Equation (5) that the key to the time scale algorithm is the weight $\omega_{i}$ and the estimated value of the frequency difference ${\hat{y}}_{\mathrm{ie}}$. The estimated values ${\hat{y}}_{\mathrm{ie}}$ used in the KPW algorithm are calculated by Kalman filtering. Since the Kalman filter algorithm can effectively remove frequency white noise, frequency random walk noise and frequency random run noise, it can improve the frequency stability of the time scale algorithm output. The five basic equations of the Kalman filtering algorithm [7] are:(6)$$\begin{array}{l} \left\{ \begin{array}{l} {\hat{X}}_{k,k-1}=AX_{k-1,k-1}\;\;\;\;\;\;\;\;\;\;\;\;\;\;\;\;\;\;\;\;\;\;\;\;\;\;\;\;\;\;\;\;\;\;(a)\\ P_{k,k-1}=AP_{k-1,k-1}A^{\prime }+Q\;\;\;\;\;\;\;\;\;\;\;\;\;\;\;\;\;\;\;\;\;\;\;\;(b)\\ K_{k}=P_{k,k-1}H^{\prime }{\left[HP_{k,k-1}H^{\prime }+R\right]}^{-1}\;\;\;\;\;\;\;(c)\\ {\hat{X}}_{k,k}={\hat{X}}_{k,k-1}+K_{k}[Z_{k}-H{\hat{X}}_{k,k-1}]\;\;\;\;\;(d)\\ P_{k,k}=(I-K_{k}H)P_{k,k-1}\;\;\;\;\;\;\;\;\;(e) \end{array}\right. \end{array}$$The Kalman filtering algorithm is a consensus in related fields, and this article will not introduce the parameters in detail [7]. The following mainly introduces the specific parameter determination method in this paper. The system model of the atomic clock can be represented by a three-dimensional linear discrete system:(7)$$\left[\begin{array}{l} x(t+\Delta t)\\ y(t+\Delta t)\\ z(t+\Delta t) \end{array}\right]=\left[\begin{array}{ccc} 1 & t & \frac{1}{2}t^{2}\\ 0 & 1 & t\\ 0 & 0 & 1 \end{array}\right].\left[\begin{array}{l} x(t)\\ y(t)\\ z(t) \end{array}\right]+\left[\begin{array}{l} \Delta x\\ \Delta y\\ \Delta z \end{array}\right]$$
where x(t), y(t), z(t) represent the three deterministic trend items of clock difference, frequency difference and frequency drift, respectively. Δx, Δy, Δz represent the errors caused by phase noise and frequency noise independent of the deterministic trend item, respectively. Its noise matrix is:(8)$$\begin{array}{l} W_{}=\left[\begin{array}{l} \Delta x\\ \Delta y\\ \Delta z \end{array}\right] \end{array}$$The process noise covariance q of the Kalman filter is equal to the covariance matrix W [7], as shown in Equation (9)
(9)$$q_{}=E[W_{}{W^{\prime }}_{}]=\left[\begin{array}{ccc} q_{1}t+q_{2}t^{3}/3+q_{3}t^{5}/20 & q_{2}t^{2}/2+q_{3}t^{4}/8 & q_{3}t^{3}/6\\ q_{2}t^{2}/2+q_{3}t^{4}/8 & q_{2}t+q_{3}t^{3}/3 & q_{3}t^{2}/2\\ q_{3}t^{3}/6 & q_{3}t^{2}/2 & q_{3}t \end{array}\right]$$
where $q_{1}$, $q_{2}$ and $q_{3}$ represent the process noise parameters of White Frequency Modulation (WFM), Random Walk Frequency Modulation (RWFM) and Random Run Frequency Modulation (RRFM), respectively, which are estimated by least squares fitting of Hadamard total variance [24]:(10)$$Total\sigma_{H}^{2}(\tau )=\left(10/3\right)q_{0}\tau^{-2}+q_{1}\tau^{-1}+q_{2}\tau /6+11q_{3}\tau^{3}/120$$

Through Equation (5), we can see that another key factor of the KPW algorithm is the weight. The weight selection equation of the traditional weighted time scale algorithm is generally [8]:(11)$$\omega_{i}={\left[\sum_{i=1}^{N}\frac{1}{\sigma_{H}(\tau )}\right]}^{-1}[\frac{1}{\sigma_{H}(\tau )}]$$
where $\sigma_{H}^{2}(\tau )$ represents the Hadamard variance when the sampling interval is τ, τ represents the sampling interval, and N represents the number of atomic clocks in the clock group. Since the Hadamard variance is a three-sample variance, it is not sensitive to linear frequency drift, so it is suitable for analyzing the frequency stability of rubidium atomic clocks. The expression for Hadamard variance [25] is:(12)$$\sigma_{H}^{2}(\tau )=\frac{1}{6(N-3)\tau^{2}}\sum_{i=1}^{N-3}{\left[x_{i+3}-3x_{i+2}+3x_{i+1}-x_{i}\right]}^{2}$$

### 3.2. Genetic Algorithm

Genetic Algorithm (GA) was first proposed by John Holland in the United States in the 1970s [10]. This algorithm is designed and proposed according to the evolution law of organisms in nature. It is a computational model of the biological evolution process that simulates natural selection and genetics mechanisms, and a method of searching for optimal solutions by simulating the natural evolution process. GA expresses the decision variables in the problem space as an individual in the genetic space through a certain coding method, which is a genotype string structure data. At the same time, the value calculated by the fitness function is used to evaluate the quality of the individual and serve as the basis for genetic operations. Genetic operations include the following three basic operators (genetic operator): selection; crossover; mutation. The basic process of genetic algorithm is shown in the Figure 2:

Through the investigation of the literature, it is found that each time scale algorithm either adopts an equal weight method for the clock groups composed of the same type of clocks, or uses the Equation (11) to determine the weight [5,8,26]. When the constellation runs autonomously for a long time, the performance of the atomic clock will change with time, so the method of equal weight is not applicable to the actual situation. When Equation (11) is used to determine the weight $\omega_{i}$, the weight changes with the change of the sampling interval, and the final generated time scale also changes with the change of the sampling interval. In this paper, the genetic algorithm is used to optimize τ in Equation (11); the problem space of the GA represents the sampling interval τ in Equation (11). Then, the calculation model of the genetic algorithm combined with KPW time scale algorithm is:(13)$$\{{}_{\begin{array}{l} f(P)=\sigma_{TA(P)}(\tau ) \end{array}}^{TA(P)=\sum_{i=1}^{N}{\left[\sum_{m=1}^{N}\frac{1}{\sigma_{m}(P)}\right]}^{-1}[\frac{1}{\sigma_{i}(P)}](x_{ij}(t)-({\overset{}{x}}_{ie}(t-\Delta t)+\Delta t\overset{}{{\hat{y}}_{ie}(t-\Delta t)+1/2D{\left(t-\Delta t\right)}^{2})}}$$where TA(P) is the time scale generated by calculating the problem space using the KPW time scale algorithm in Section 3.1, $\sigma_{TA(P)}(\tau )$ is the Hadamard variance of this time scale, and τ is the sampling interval corresponding to the Hadamard variance. The initial parameters of the GA in this paper are respectively set as follows: the problem space P is coded in decimal, the length L of the coded string of each variable is 4, the population size M is 50, and the termination algebra T is 50. The genetic operator adopts the crossover operator and mutation operator, respectively, and the crossover probability and mutation probability are set to 0.9 and 0.2, respectively.

## 4. Time-Frequency Driving Algorithm Principle

### 4.1. Atomic Frequency Standard Error Model

Frequency stability is an important indicator of atomic frequency standards, which refers to the frequency or phase changes within a specified time interval. The main causes include deterministic trend items and non-deterministic noise items. The error mathematical model of atomic frequency standards [15] is shown in Equation (14):(14)$$x\left(t\right)=x_{0}+y_{0}t+\frac{Dt^{2}}{2}+\varepsilon \left(t\right)$$
where the deterministic trend term includes the initial time error $x_{0}$, the initial frequency deviation $y_{0}$, and the linear frequency drift D caused by aging and temperature changes. ε(t) is the random frequency deviation of different types of noise such as stationary noise and non-stationary noise, mainly including: WPM, FPM, WFM, FFM and RWFM noise, and it can be represented by the power law noise spectrum model of the classic oscillator given by Lesson [27]:(15)$$S_{y}\left(f\right)=h_{2}f^{2}+h_{1}f^{1}+h_{0}+h_{-1}f^{-1}+h_{-2}f^{-2}$$
where $S_{y}\left(f\right)$ represent the power spectral density; $h_{2}$, $h_{1}$, $h_{0}$, $h_{-1}$ and $h_{-2}$ represent the noise term coefficients of WPM, Flicker Phase Modulation (FPM), WFM, Flicker Frequency Modulation (FFM) and RWFM, respectively.
(16)$$S_{y}\left(f\right)=\frac{f^{2}}{f_{0}^{2}}S_{\phi }\left(f\right)=2\frac{f^{2}}{f_{0}^{2}}L\left(f\right)$$
where $S_{\phi }\left(f\right)$ is the phase noise spectral density, $f_{0}$ is the center frequency of the frequency source, and f is the sideband frequency. From Equations (15) and (16), the SSB phase noise spectral density of the frequency standard [28] can be obtained per Equation (17):(17)$$L\left(f\right)=\frac{f_{0}^{2}}{2}\left(h_{-2}f^{-4}+h_{-1}f^{-3}+h_{0}f^{-2}+h_{1}f^{-1}+h_{2}\right)$$Five kinds of noise parameters can be fitted by non-negative least squares for total Hadamard variance.

### 4.2. Algorithm Principle of Equivalent DPLL Steering Control System

The time-frequency steering system adopts the Kalman filter shown in Equation (6), and the three-dimensional state-space model of phase difference, frequency difference and frequency drift established is [19,22]:(18)$$\{\begin{array}{c} X_{k+1}=A\cdot X_{k}+J_{k}\\ Z_{k}=H\cdot X_{k}+w_{k} \end{array}$$In Equation (18), the first equation is the state equation, $X_{k}={\left[\begin{array}{ccc} x_{k} & y_{k} & d_{k} \end{array}\right]}^{T}$, $x_{k}$, $y_{k}$ and $d_{k}$ respectively represent the three state variables of phase, frequency, and frequency change rate. $J_{k}={\left[\begin{array}{ccc} 0 & 0 & u_{k} \end{array}\right]}^{T}$, $u_{k}$ is process noise.
(19)$$A=\left[\begin{array}{ccc} 1 & T & \frac{1}{2}\cdot T^{2}\\ 0 & 1 & T\\ 0 & 0 & 1 \end{array}\right]$$
where A is the state transition matrix, T is the sampling time interval. The second equation in Equation (12) is the measurement equation, $Z_{k}$ is the measured value, $w_{k}$ is the measurement noise, $H=\left[\begin{array}{ccc} 1 & 0 & 0 \end{array}\right]$ is a linear connection matrix, which describes the phase deviation process between the to-be-driven frequency standard and the driving reference frequency standard in the time-frequency steering system. The covariance of the process noise $u_{k}$ is:(20)$$Q=E\left[J_{k}\cdot J_{k}^{T}\right]=\left[\begin{array}{ccc} 0 & 0 & 0\\ 0 & 0 & 0\\ 0 & 0 & E\left[u_{k}^{2}\right] \end{array}\right]=\left[\begin{array}{ccc} 0 & 0 & 0\\ 0 & 0 & 0\\ 0 & 0 & Q_{33} \end{array}\right]$$
where $u_{k}\sim N\left(0,Q_{33}\right)$. The covariance of measurement noise $w_{k}$ is $R=E\left[w_{k}^{2}\right]$, $w_{k}\sim N\left(0,R\right)$. Equations (21) and (22) represent the input–output relationship between the measured value $Z_{k}$ and the estimated value ${\hat{x}}_{k}$ of the phase state variable in the steady-state Kalman filter.
(21)$$\tilde{X}=\frac{G^{\prime }\left(z\right)}{1-K_{s1}+G^{\prime }\left(z\right)}\cdot \tilde{Z}$$
(22)$$G^{\prime }\left(z\right)=\frac{Ks_{1}\cdot {\left(1-z^{-1}\right)}^{2}+\left(Ks_{2}\cdot T+\frac{1}{2}\cdot Ks_{3}\cdot T^{2}\right)\cdot z^{-1}\cdot \left(1-z^{-1}\right)+Ks_{3}\cdot T^{2}\cdot z^{-2}}{{\left(1-z^{-1}\right)}^{3}}$$
where $\tilde{X},\tilde{Z}$ are the z-transform forms of ${\hat{x}}_{k}$ and $Z_{k}$, respectively. Through Equation (21), the closed-loop system transfer function structure consistent with the third-order type-three DPLL can be obtained [23], which is a further extension of the two-dimensional state-space model Kalman filter [22]. In order to ensure the normal and orderly output of the time-frequency steer value in the loop filter, it is considered to add a delay device $z^{-1}$. From Equation (21) and the delay device, the open-loop system transfer function G(z), closed-loop system transfer function H(z) and closed-loop error transfer function He(z) equivalent to the DPLL system can be obtained, which are expressed as Equations (23)–(25):(23)$$G\left(z\right)=\frac{z^{-1}}{1-K_{s11}}\cdot G^{\prime }\left(z\right)$$
(24)$$H\left(z\right)=\frac{G\left(z\right)}{1+G\left(z\right)}$$
(25)$$He\left(z\right)=\frac{1}{1+G\left(z\right)}$$

The Kalman filter guarantees the stability of the system because of its complete observability, and the equivalent DPLL with a delay is represented by Equations (23) and (24); in addition, (25) has also been verified by simulation. The noise variance parameters $Q_{22}$ and R are only used to determine the steady-state Kalman gain $K_{s}$ equivalent to the DPLL gain when constructing the equivalent DPLL, WU Yi-wei [22] derived and verified the approximate relationship between the noise variance parameters $Q_{22}$, R and the steady-state Kalman gain $K_{s}$ in the two-dimensional Kalman filter model. By analyzing the five steps of Kalman filtering, $P_{k,k}$ is the estimated covariance matrix, and $P_{k,k-1}$ is the predicted covariance matrix. When the Kalman filter reaches a steady state, $P_{k,k}$ and $P_{k,k-1}$ can converge to the steady state values Ps and $Ps^{-}$, respectively, and the Equation (6b) can be rewritten as
(26)$$\left(\begin{array}{c} Ps_{11}^{-}\\ Ps_{12}^{-}\\ Ps_{13}^{-}\\ Ps_{22}^{-}\\ Ps_{23}^{-}\\ Ps_{33}^{-} \end{array}\right)=\left(\begin{array}{cccccc} 1 & 2T & T^{2} & T^{2} & T^{3} & \frac{1}{4}T^{4}\\ 0 & 1 & T & T & \frac{3}{2}T^{2} & \frac{1}{2}T^{3}\\ 0 & 0 & 1 & 0 & T & \frac{1}{2}T^{2}\\ 0 & 0 & 0 & 1 & 2T & T^{2}\\ 0 & 0 & 0 & 0 & 1 & T\\ 0 & 0 & 0 & 0 & 0 & 1 \end{array}\right)\left(\begin{array}{c} Ps_{11}\\ Ps_{12}\\ Ps_{13}\\ Ps_{22}\\ Ps_{23}\\ Ps_{33} \end{array}\right)+\left(\begin{array}{c} 0\\ 0\\ 0\\ 0\\ 0\\ Q_{33} \end{array}\right)$$Equation (6c) can be rewritten as
(27)$$Ks_{i}=\frac{Ps_{1i}^{-}}{Ps_{11}^{-}+R}(i=1,2,3)$$Through Equation (27), Equation (6e) can be rewritten as
(28)$$Ps_{1i}^{}=Ks_{i}\cdot R=\frac{R\cdot Ps_{1i}^{-}}{Ps_{11}^{-}+R}\left(i=1,2,3\right)$$
(29)$$Ps_{2i}^{}=Ps_{2i}^{-}-\frac{Ps_{12}^{-}\cdot Ps_{1i}^{-}}{Ps_{11}^{-}+R}\left(i=2,3\right)$$
(30)$$Ps_{33}^{}=Ps_{33}^{-}-\frac{{\left(Ps_{13}^{-}\right)}^{2}}{Ps_{11}^{-}+R}$$It can be known from experiments that $Ps_{11}^{-}<<R$. From Equations (26), (27) and (30), we can get
(31)$$Ks_{3}\approx \sqrt{\frac{Q_{33}}{R}}$$Combining Equations (26), (27) and (29) to get
(32)$$Ks_{2}=\sqrt{2\cdot Ks_{1}\cdot Ks_{3}}$$Combining Equations (26)–(29) and (32) to get
(33)$$Ps_{11}^{-}=\frac{2Ks_{2}T}{Ks_{1}-Ks_{2}T+\frac{1}{2}Ks_{3}T^{2}}R$$It can be known from experiments that $Ks_{2}\cdot T<<Ks_{1}$, $\frac{1}{2}\cdot Ks_{3}\cdot T^{2}<<Ks_{1}$. Combine Equations (27) and (31)–(33) to get
(34)$$Ks_{2}\approx 2\cdot \sqrt[{3}]{T\cdot Ks_{3}{}^{2}}\approx 2\cdot \sqrt[{3}]{T\cdot \frac{Q_{33}}{R}}$$
(35)$$Ks_{1}\approx \sqrt{2T}\cdot \sqrt{Ks_{2}}\approx 2\cdot \sqrt[{6}]{T^{4}\cdot \frac{Q_{33}}{R}}$$From Equations (31), (34) and (35), it can be seen that if T has been determined, the noise variance parameter can be proportionally composed of parameter $\frac{R}{Q_{33}}$ to calculate the DPLL gain. The final Equations (24) and (25) completely determine the performance of the steering control system of the equivalent DPLL. The functional block diagram of the equivalent third-order DPLL steering control system is shown in Figure 3:

Equation (36) can be obtained from the Figure 3:(36)$$z\cdot TA_{steered}\left(Rb\right)=G\left(z\right)\cdot \left(TA\left(Cs\right)-TA_{steered}\left(Rb\right)\right)+TA\left(Rb\right)$$
where TA(Cs) is the driving reference, TA(Rb) is to be driven, and $TA_{steered}(Rb)$ is TA(Rb) after being steered. From Equations (23), (24) and (36), we can get:(37)$$z\cdot TA_{steered}\left(Rb\right)-TA\left(Rb\right)=G\left(z\right)\cdot Err\left(z\right)=\frac{1}{\left(1-Ks_{1}\right)}\cdot \left(Ks_{1}\cdot \frac{z^{-1}}{1-z^{-1}}+\left(Ks_{2}+\frac{1}{2}Ks_{3}\cdot T\right)\cdot {\left(\frac{T\cdot z^{-1}}{1-z^{-1}}\right)}^{2}+Ks_{3}\cdot {\left(\frac{T\cdot z^{-1}}{1-z^{-1}}\right)}^{3}\right)\cdot Err\left(z\right)$$
where $Err\left(z\right)=TA\left(\mathrm{Cs}\right)-TA_{steered}\left(Rb\right)$ is the steering error of TA(Cs) and $TA_{steered}(Rb)$. After Equation (37) is inversely transformed, the control value Equation (38) for each sampling moment i of the TA(Rb) and the control reference TA(Cs) in the time domain can be obtained:(38)$$\frac{1}{\left(1-Ks_{1}\right)}\cdot \sum_{j=1}^{i}\left(Ks_{1}\cdot Err\left(j\right)+\sum_{k=1}^{j-1}T\cdot \left(\left(Ks_{2}+\frac{1}{2}Ks_{3}\cdot T\right)\cdot Err\left(k\right)+\sum_{l=1}^{k-1}T\cdot Ks_{3}\cdot Err\left(l\right)\right)\right)$$

### 4.3. Using GA to Improve Control System Parameter Selection

Use Equation (17) to draw the SSB phase noise spectral density curves $L{\left(f\right)}_{TA(Cs)}$ and He(f), use the frequency value of the intersection point of the two curves as a fixed reference value f(L(f)). Then, substituting $z=e^{j2\pi fT}$ into Equations (24) and (25), the amplitude–frequency response curves H(f) and He(f) of the system transfer function H(z) and system error transfer function He(z) in the frequency domain can be obtained [22].

H(f) and He(f) respectively have the characteristics of low-pass and high-pass filters in the frequency domain, and the frequency intersection value of the two frequency response curves is used as the dynamic debugging value f(H). Based on the principle of optimal overall frequency stability, when the output signal of the driving system fully takes into account the frequency stability advantages of TA(Cs) and TA(Rb), it is necessary to make the dynamic tuning value f(H) consistent with the fixed reference value f(L(f)) as much as possible. However, the existing parameter adjustment of the equivalent DPLL steering control system requires multiple discrete debugging and approximate selection, so there are precision limitations and adaptive reliability problems, thus further improvement is necessary. In this section, a genetic algorithm is introduced to improve the selection process of noise variance parameter pairs. The driving system expects the overall frequency stability of the output signal to be better, so the more consistent the dynamic debugging value f(H) is with the fixed reference value f(L(f)), the better. The fitness function to establish the minimization of the deviation of the control intersection frequency value is as follows:(39)Fcost=|f(H)−f(L(f))|α=|ffdownfupind(min|H(f)−He(f)|)α−ffdownfupind(min|L(f)TA(Cs)−L(f)TA(Rb)|)α|
where || means to obtain the absolute value of the deviation, ffdownfupind(min||)α means to search for the corresponding frequency value at the intersection of two frequency domain curves with the minimum resolution a within the frequency search threshold given in $\left[f_{down},f_{up}\right]$.

The genetic algorithm in this section adopts the decimal coding method. The length *L* of the coding string of each variable is 10, the population size *M* is 100, and the termination algebra *T* is 100. Genetic operators that use crossover and mutation, $P_{c}$ and $P_{m}$, are 0.9 and 0.2, respectively.

## 5. Simulation Results and Analysis

### 5.1. Experimental Analysis of Time Scale Algorithm

The experimental data in this paper used the 300-s precision clock difference data provided by the Center for Orbit Determination in Europe (CODE), and the data collection time is from 2220 weeks to 2226 weeks of GPS. Satellites G1, G3, G4, G6, G9, G11, G14, G18, G24, G27, G30 are rubidium clocks; satellite numbers are G2, G5, G7, G12, G15, G16, G17, G19, G20, G21, G22, G29, G31 are cesium clocks. In this paper, clocks are divided into two clock groups according to their types: rubidium clock group and cesium clock group. The two clock groups use the KPW time scale algorithm respectively, and use Equation (11) to assign weights. The two time scales obtained are TA(Rb) and TA(Cs), respectively, and the frequency stability of the single clocks in their respective clock groups are shown in Figure 4 and Figure 5.

It can be seen from Figure 4 and Figure 5 that the stability of TA(Rb) and TA(Cs) is better than any single clock in their respective clock groups.

Next, the value range of τ in Equation (11) is [300, 1.5 × 10^5^], and 10 values are randomly selected within this range. The two clock groups use the KPW algorithm to calculate seven time scales respectively. Their stability is as follows:

It can be seen from Figure 6 that when using the KPW algorithm, with the change of 1, the stability of the two output time scales TA(Rb) and TA(Cs) will also change. When the sampling interval is less than 8 × 10^4^, TA(Rb) is better in stability than TA(Cs), and when the sampling interval is greater than 8 × 10^4^, the stability of TA(Rb) is worse than TA(Cs).

This paper intends to adopt the time-frequency steering algorithm, using TA(Cs) to drive TA(Rb), the short-term stability and long-term stability of the time scale after driving depend on TA(Rb) and TA(Cs), respectively. Therefore, the genetic algorithm is used to optimize the sampling interval τ, so that the stability of TA(Rb) is optimal before 8 × 10^4^, and the stability of TA(Cs) is optimal after 8 × 10^4^ s. In the end, the overall stability of the time-frequency steering algorithm is a relatively good time scale. Let the fitness function of the GA of the rubidium clock group be $f(P)=\sigma_{TA(P)}(2\times 10^{4})$, the fitness function of the genetic algorithm of the cesium clock group be $f(P)=\sigma_{TA(P)}(5\times 10^{5})$, and the optimal range of both be $\left[\;300∼1.5\times 10^{5}\right]$. The results of the genetic algorithm simulation are as follows:

The red point in Figure 7 is the minimum fitness obtained by the genetic algorithm, and the two points correspond to 9300 and 132,659, respectively. The two red curves in Figure 7 are the stability of the time scale calculated using the minimum fitness value. It can be seen from the left figure of Figure 8 that when τ=9300, the frequency stability of TA(Rb) is the smallest in the interval [300–8 × 10^4^], and it can be seen from the right figure of Figure 8 that when τ=132,659, the frequency stability of TA(Cs) is the smallest in the interval [8 × 10^4^–1 × 10^6^].

Next, use the improved time-frequency steering algorithm in Section 4 for time-frequency steering, let TA(Cs) be used as the steering reference, and TA(Rb) be used as the signal to be steered. The specific simulation results are introduced in detail in Section 5.2.

### 5.2. Experimental Analysis of Time-Frequency Steering Algorithm

According to the physical parameters of the time scale TA(Rb) and TA(Cs), the fitness function for optimal parameter selection introduced in the previous section can be constructed through Equations (17), (24), (25) and (41).The SSB phase noise spectral densities of TA(Cs) and TA(Rb) are $L{\left(f\right)}_{TA\left(Cs\right)}$ and $L{\left(f\right)}_{TA\left(Rb\right)}$; the intersection point of $L{\left(f\right)}_{TA\left(Cs\right)}$ and $L{\left(f\right)}_{TA\left(Rb\right)}$ is $f\left(L\left(f\right)\right)=5.4236\times 10^{-6}(\mathrm{Hz})$. Therefore, the optimization range of the genetic algorithm is $\left[f_{down},f_{up}\right]=\left[5.4\times 10^{-6},5.5\times 10^{-6}\right](\mathrm{Hz})$. The minimum resolution for optimization is $1\times 10^{-11}\mathrm{Hz}$. Normalize the noise variance parameter R to $Q_{33}$; the genetic algorithm iteratively adjusts the normalized R and can approach the minimum value of the fitness function. The iterative convergence of the optimal fitness value is shown in Figure 9.

The selection range of parameter R/$Q_{33}$ in Figure 10 is $\left[10^{20},10^{30}\right]$. The green mark represents the fitness value corresponding to the initial value of the population, and the red circle represents the minimum value of the fitness function obtained by the genetic algorithm. Because the minimum resolution of the fitness function is preset, when the emergence of the optimal population minimizes the fitness function value to 0, it indicates that the deviation between the dynamic debugging value f(H) and the fixed reference value f(L(f)) exceeds the expected minimum resolution.

Using time scale TA(Cs) and TA(Rb) data, select a group of optimal noise variance parameters obtained by genetic algorithm as group 1, make two types of intersection points in the frequency domain, and make statistics on the frequency deviation of the two types of intersection points, compared with the parameter groups 2–5 under the time-frequency steering mode of the existing discrete debugging and approximate parameter selection; the comparison results are shown in Figure 10 and Table 1. Obviously, the improved parameter selection method combined with the genetic algorithm has better deviation accuracy. Substituting the optimal DPLL gain into the closed-loop system transfer function (24), we get:$$H\left(z\right)=\frac{0.0101z^{2}-0.0201z+0.01}{0.9899z^{3}-2.9596z^{2}+2.9496z-0.9799}$$

We can solve a set of poles, 0.9975±0.0043i and 0.9949, because two conjugate complex poles and one real pole are in the unit circle of the Z plane; the DPLL system determined by the optimal DPLL gain is stable.

Figure 11 and Figure 12 have shown the frequency stability and steering error of the output signal after being driven under different parameter selection situations. Parameter group 1 $TA{\left(Rb\right)}_{1}$ is the time scale output by the time-frequency steering algorithm optimized by the genetic algorithm. Its frequency stability does not appear to be biased and unbalanced, and it can better take advantage of the frequency stability of each sampling interval period and its time. The steering error is in the middle of the balance. However, the frequency stability swing of the output signal corresponding to the parameter groups 2–4 corresponding to the existing manual repeated debugging method is obviously larger, and is the best overall under the balance of advantages that cannot be more accurately understood.

Figure 10, Figure 11 and Figure 12 and Table 1 also jointly characterize the influence of the time-frequency steer value determined by the five parameter control groups on the driving effect. When the noise variance parameter group is normalized by $Q_{33}$, the larger R will affect the dynamic debugging value f(H) and deviate from the fixed reference value f(H) to the left. At this time, the time-frequency steer value determined by the parameter group will expand the influence of the steered frequency standard TA(Rb), so that the signal output after being steered is biased towards the improvement of short-term stability performance, which unbalances the performance of long-term stability and increase the time control error. On the contrary, the change of the corresponding result is the opposite.

Let TA(Cs) be used as the steering reference, TA(Rb) be used as the frequency standard to be steered; use the genetic algorithm to optimize the selection of steering parameters, and the time scale output after steering is TA(Streed).

Figure 13 shows the phase differences of the three time scales. When the steering enters a stable state, the phase of TA(Streed) is synchronized with the steering reference TA(Cs), and the error is within 0.5 ns. Figure 14 shows the stability of the three time scales. It can be seen that the frequency stability of TA(Streed) is consistent with TA(Rb) when the sampling interval is less than 1 × 10^4^ s, and is consistent with TA(Cs) when it is greater than 2 × 10^5^ s. Therefore, the time-frequency signal output by the improved time-frequency steering algorithm in this paper can better comprehensively control the frequency stability advantages of the reference and the steered frequency standard.

Using the traditional KPW time scale algorithm, 24 clocks are formed into a clock group to output the time scale TA(KPW), and the time scale output by the method proposed in this paper is TA(Steered). Their frequency stability comparison is shown in Figure 15:

It can be seen from Figure 15 that the frequency stability of TA(Steered) in this paper is better overall. When the sampling intervals are 300 s, 1 day, and 1 × 10^−7^ s, its frequency stability is 5.73 × 10^−14^, 9.17 × 10^−15^, 4.93 × 10^−16^. Compared with the frequency stability of TA(KPW), when the sampling interval is 300 s, the frequency stability is improved by 71%; when the sampling interval is 1 day, the frequency stability of the two is equivalent; when the sampling interval is 1 × 10^6^ s, the frequency stability is increased by 94%. Compared with the traditional KPW time-scale algorithm, the method proposed in this paper has significantly improved the short-term and long-term stability of the time-frequency signal output.

The frequency stability comparison between TA(Steered) and 24 single clocks is shown in Figure 16 and Table 2:

From Figure 16 and Table 2, it can be seen that TA(Steered) has a higher overall advantage in stability than a single atomic clock in each period. When the sampling interval is 300 s, it can reach the order of 5.73 × 10^−14^, the stability of 1000 s can reach 2.79 × 10^−14^, the stability of 1 × 10^5^ s can reach 8.38 × 10^−15^, and the level of 1 × 10^6^ s can reach 8.87 × 10^−16^. When the sampling interval is 1 × 10^6^ s, it is six times more stable than the best cesium clock and three orders of magnitude better than the worst rubidium clock. After analyzing the stability, the next step is to analyze the comprehensive atomic time TA(Steered) and the forecast clock difference of each atomic clock. The results are shown in Figure 17and Table 3 below:

From Figure 17 and Table 3, it can be seen that the time scale TA(Streed) forecast has the smallest forecast residual for ten days, which can reach the level of 4.57 × 10^−9^ s, which is two orders of magnitude higher than that of the rubidium clock G6 with the largest forecast residual. The clock error prediction error of a single atomic clock gradually increases with time, and can reach the order of 400 ns in ten days. Therefore, it can be seen that the time scale generation algorithm proposed in this paper can effectively improve the frequency stability of the output signal and improve its predictability.

## 6. Conclusions

This paper proposes a navigation constellation-level high-precision time scale generation method, presents the navigation constellation time scale generation framework, and designs an improved KPW time scale algorithm and time-frequency control algorithm that integrates genetic algorithms. Finally, the algorithm simulation of the establishment of the constellation time scale is carried out using the GPS precision clock difference data provided by CODE. The results show that the phase deviation of time-frequency steer is kept within ±0.2 ns, and when the sampling interval is 300 s, 1 day, and 1 × 10^6^ s, its frequency stability is 5.73 × 10^−14^, 9.17 × 10^−15^, and 8.87 × 10^−16^, respectively. Compared with the traditional KPW time scale algorithm, the short-term (300 s) and long-term (1 × 10^6^ s) stability have increased by 71% and 94%, respectively, which have been greatly improved. The constellation-level high-precision time scale generation method proposed in this paper can significantly improve the stability performance of long-term autonomous timekeeping of navigation constellations, which is of great significance to the research of long-term autonomous timekeeping of navigation constellations based on inter-satellite links.

## Figures and Tables

**Figure 1 sensors-23-01703-f001:**
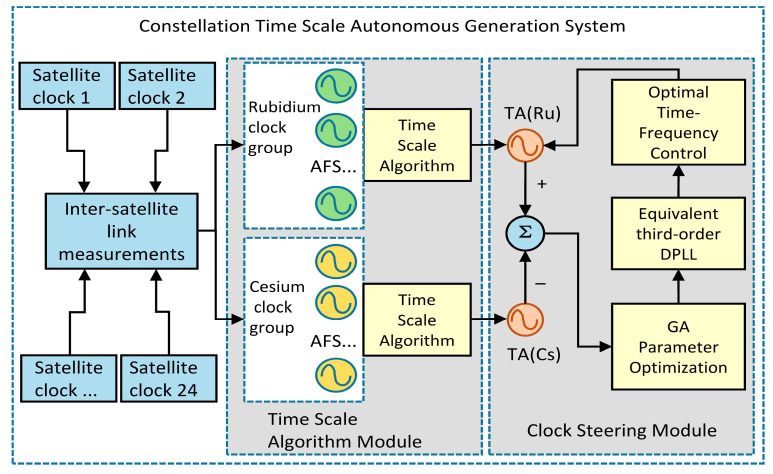
Comprehensive time scale generation system architecture diagram.

**Figure 2 sensors-23-01703-f002:**
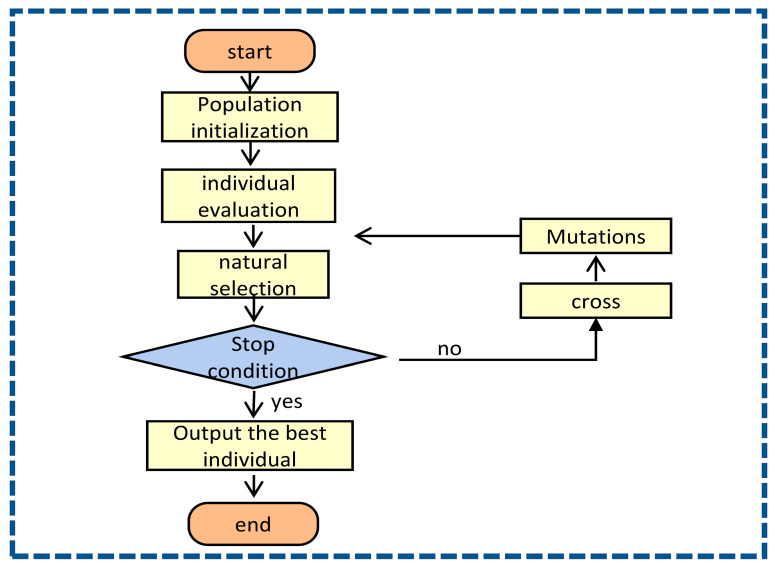
Genetic Algorithm Parameter Optimization Flowchart.

**Figure 3 sensors-23-01703-f003:**
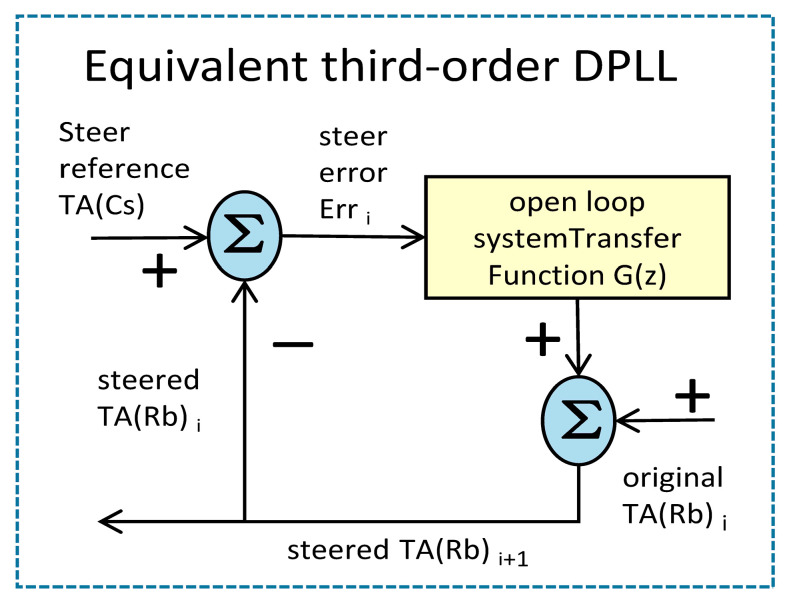
The functional block diagram of the equivalent DPLL steering control system.

**Figure 4 sensors-23-01703-f004:**
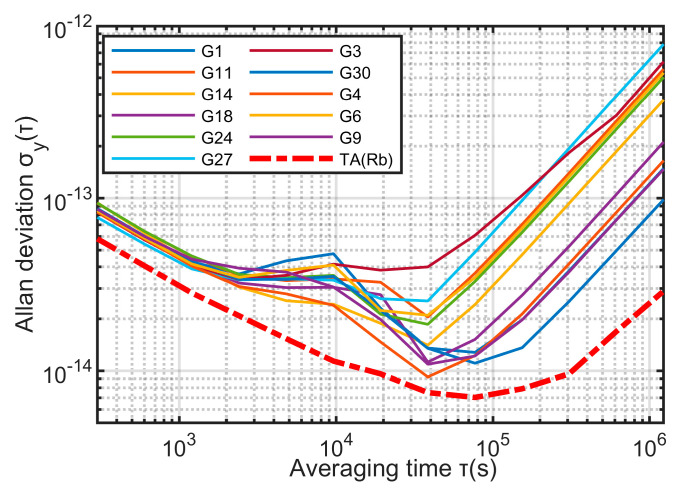
Frequency stability of rubidium clock group.

**Figure 5 sensors-23-01703-f005:**
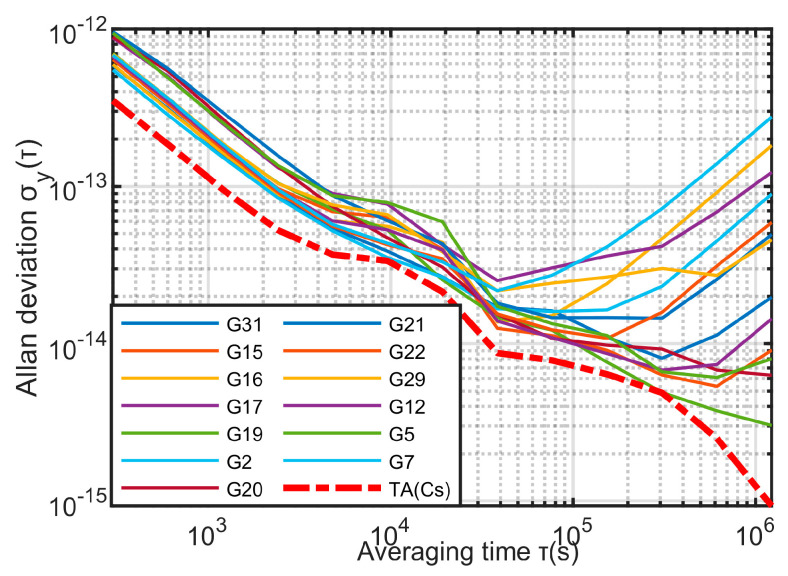
Frequency stability of cesium clock group.

**Figure 6 sensors-23-01703-f006:**
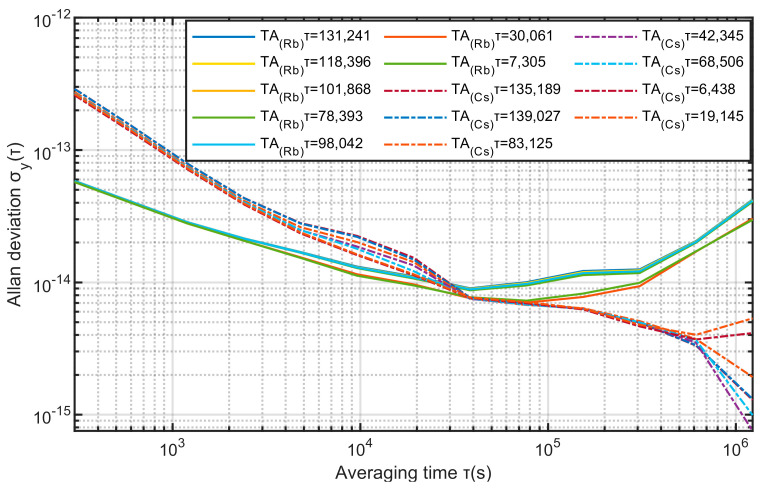
Timescales calculated for different τ.

**Figure 7 sensors-23-01703-f007:**
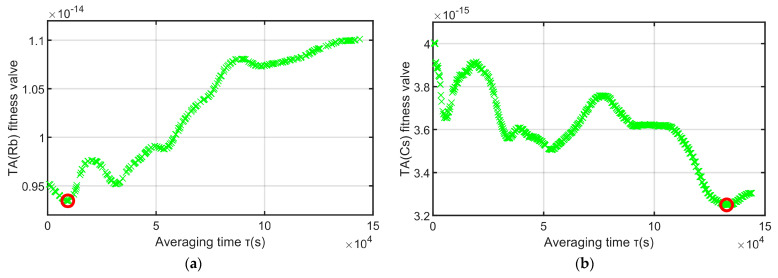
Genetic algorithm fitness.

**Figure 8 sensors-23-01703-f008:**
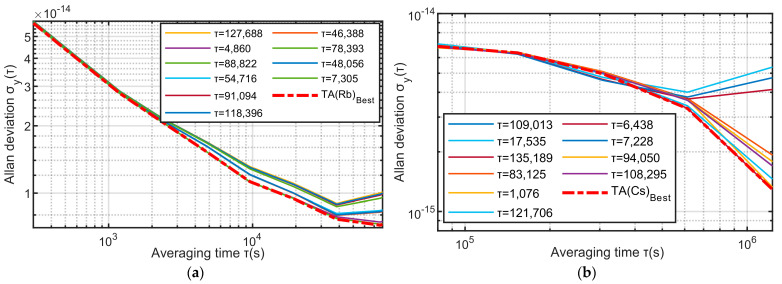
The frequency stability of the time scale of the fitness-optimal τ.

**Figure 9 sensors-23-01703-f009:**
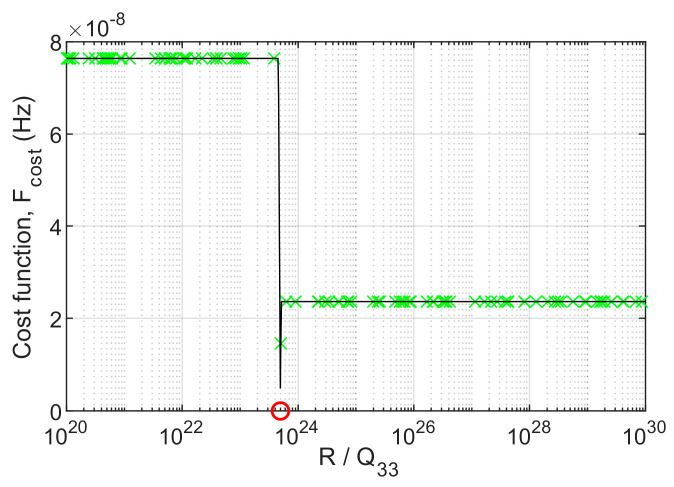
Convergence graph of genetic algorithm fitness value.

**Figure 10 sensors-23-01703-f010:**
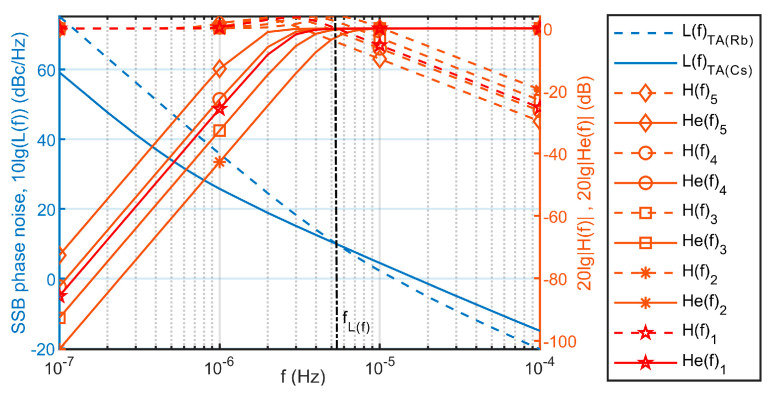
Comparison chart of frequency domain intersection points under different parameter pair selection values.

**Figure 11 sensors-23-01703-f011:**
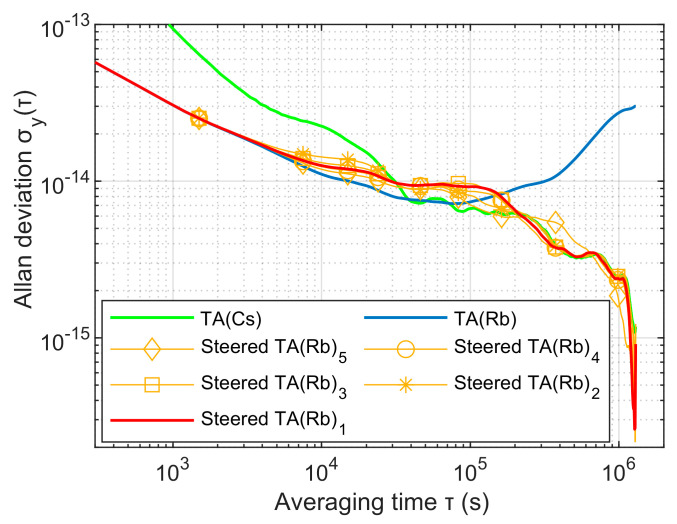
The frequency stability of steering output under the selection of different parameters.

**Figure 12 sensors-23-01703-f012:**
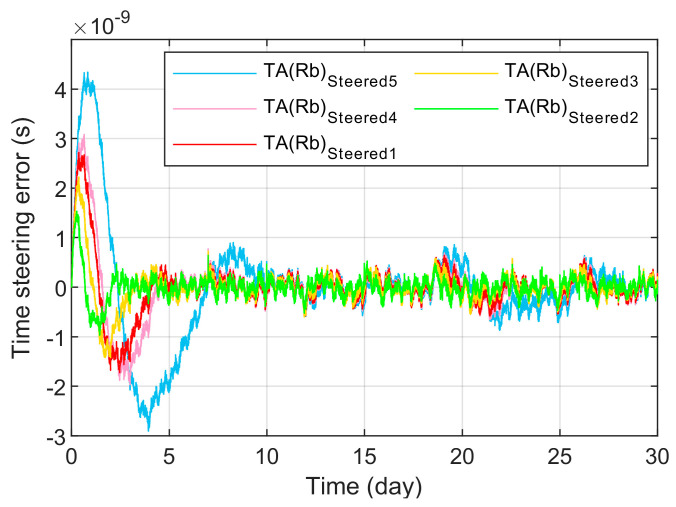
The steering error under the selection of different parameters.

**Figure 13 sensors-23-01703-f013:**
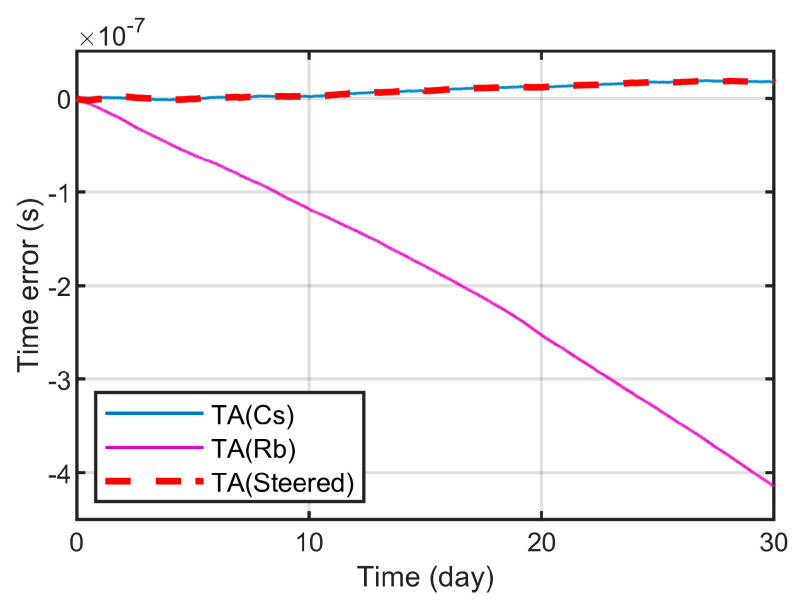
Phase difference of TA(Cs), TA(Rb), TA(Steered).

**Figure 14 sensors-23-01703-f014:**
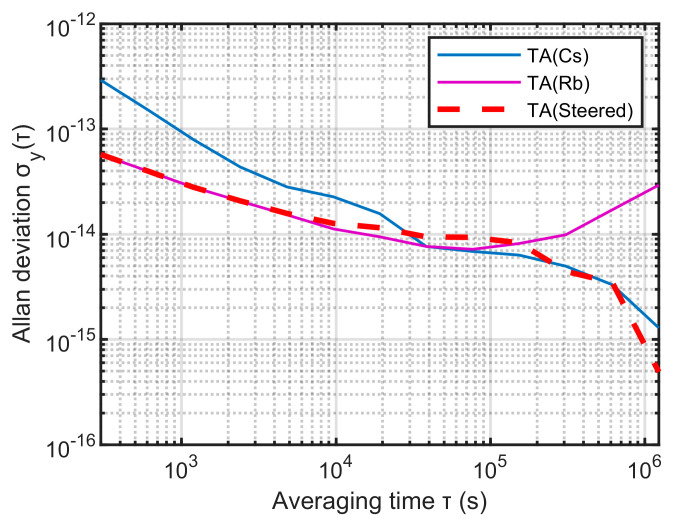
The frequency stability of TA(Cs), TA(Rb), TA(Steered).

**Figure 15 sensors-23-01703-f015:**
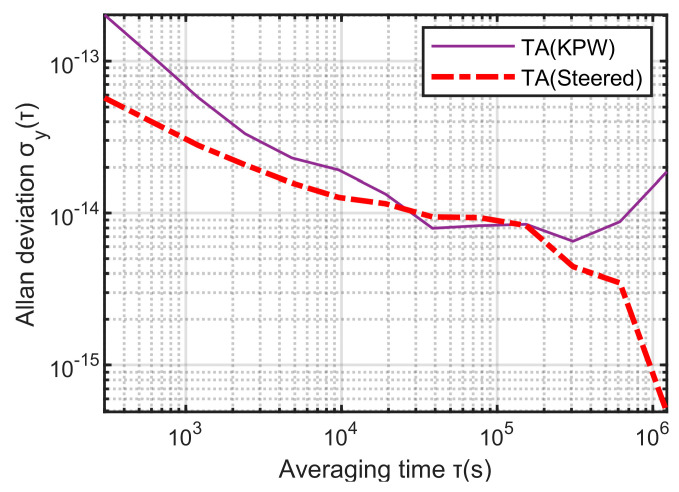
The frequency stability of TA(Steered) and TA(KPW).

**Figure 16 sensors-23-01703-f016:**
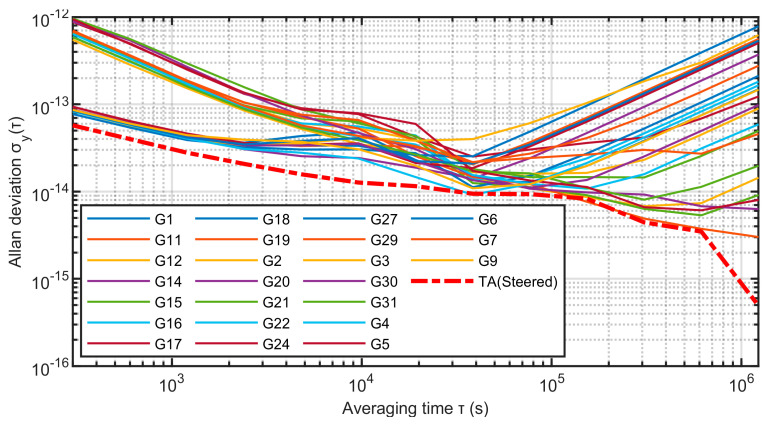
The frequency stability of TA(Steered) and 24 atomic clocks.

**Figure 17 sensors-23-01703-f017:**
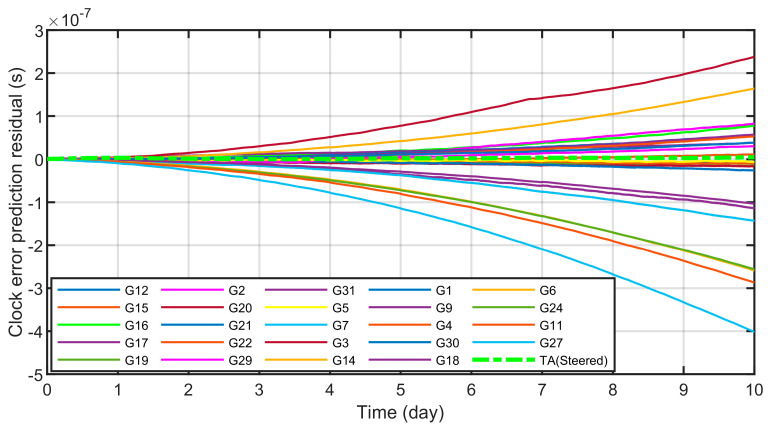
The forecast error of TA(Steered) and 24 atomic clocks.

**Table 1 sensors-23-01703-t001:** Table of cross point deviation in frequency domain under different parameter selection.

NVP Group Number	NVP Pair		DPLL Gains			FRV	DAV	Intersection Deviations (Hz)
	Q33 (s2)	R (s2)	Ks1	Ks2	Ks3	f(L(f)) (Hz)	f (H) (Hz)	
No. 1	1	4.96 × 10^23^	0.0101	1.690 × 10^−7^	1.4189 × 10^−12^	1 × 10^−5.2652^	1 × 10^−5.2657^	4.3286 × 10^−12^
No. 2	1	1 × 10^22^	0.0193	6.215 × 10^−7^	1 × 10^−11^		1 × 10^−4.9796^	5.0562 × 10^−6^
No. 3	1	1 × 10^23^	0.0132	2.884 × 10^−7^	3.1623 × 10^−12^		1 × 10^−5.1487^	1.6776 × 10^−6^
No. 4	1	1 × 10^24^	0.009	1.338 × 10^−7^	1 × 10^−12^		1 × 10^−5.3151^	5.9883 × 10^−7^
No. 5	1	1 × 10^25^	0.0061	6.214 × 10^−8^	3.1623 × 10^−13^		1 × 10^−5.4839^	2.1421 × 10^−6^

**Table 2 sensors-23-01703-t002:** Table of Frequency Stability Statistics.

Atomic Time Type	ADEV@3 × 10^2^	ADEV@1 × 10^3^	ADEV@1 × 10^5^	ADEV@1 × 10^6^
G1	8.71 × 10^−14^	4.29 × 10^−14^	1.54 × 10^−14^	1.49 × 10^−13^
G3	8.28 × 10^−14^	4.10 × 10^−14^	1.04 × 10^−13^	6.21 × 10^−13^
G4	8.29 × 10^−14^	4.05 × 10^−14^	2.13 × 10^−14^	1.62 × 10^−13^
G6	8.46 × 10^−14^	4.16 × 10^−14^	6.71 × 10^−14^	5.35 × 10^−13^
G2	5.49 × 10^−13^	1.53 × 10^−13^	1.64 × 10^−14^	8.97 × 10^−14^
G5	9.45 × 10^−14^	2.54 × 10^−13^	1.13 × 10^−14^	8.02 × 10^−15^
G20	8.89 × 10^−13^	2.68 × 10^−13^	9.78 × 10^−15^	6.30 × 10^−15^
G31	5.88 × 10^−13^	1.61 × 10^−13^	1.09 × 10^−14^	1.97 × 10^−14^
TA (Steered)	5.73 × 10^−14^	2.79 × 10^−14^	8.38 × 10^−15^	8.87 × 10^−16^

**Table 3 sensors-23-01703-t003:** Clock prediction accuracy.

Atomic Time Type	1 DAY	5 DAY	10 DAY
G1	8.35 × 10^−9^	5.03 × 10^−9^	5.70 × 10^−8^
G3	3.89 × 10^−9^	6.82 × 10^−8^	2.38 × 10^−7^
G4	1.08 × 10^−9^	7.75 × 10^−9^	5.26 × 10^−8^
G6	6.53 × 10^−9^	6.09 × 10^−8^	2.53 × 10^−7^
G2	1.77 × 10^−9^	2.77 × 10^−9^	3.03 × 10^−8^
G5	1.79 × 10^−9^	3.05 × 10^−9^	6.06 × 10^−9^
G20	2.46 × 10^−9^	7.51 × 10^−9^	2.58 × 10^−8^
G31	1.90 × 10^−9^	9.05 × 10^−9^	3.84 × 10^−8^
TA	3.05 × 10^−9^	1.57 × 10^−9^	4.57 × 10^−9^

## Data Availability

The datasets analyzed during the current study are available in the data repositories of The Crustal Dynamics Data Information System of NASA (https://cddis.nasa.gov/archive/gnss/products/ (accessed on 12 October 2022)). The datasets generated and analyzed during the current study are also available from the corresponding author upon reasonable request.

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
