# Peer review of "A Method for Autonomous Generation of High-Precision Time Scales for Navigation Constellations"

_sensors, 2023, doi:10.3390/s23031703_

Round 1
Reviewer 1 Report
The paper studies the method of autonomously generating time scales for navigational constellations.Improved Time Scale Algorithm and Time Frequency Steer Algorithm using Genetic Algorithm. It is valuable for research on future navigation systems. The paper is well written with a clear methodology and structure, and the evaluation is comprehensive, here are some suggestions:
1. What is the reference for the CODE clock difference data used in the simulation? This means that if you substitute TA(Steered) for the actual value used by CODE, the Allan deviation of the difference from CODE's reference time goes down?
2. In line 61, you mentioned that the Kalman filter mainly optimizes the long-term stability of the time scale, but its long-term forecast error will become worse. I think this statement is contradictory.
3. I noticed that the time scale algorithm applied by the Ground Punctuality Laboratory is to put different types of clocks in a clock group for calculation, your research is grouped by the type of clocks, can you explain the benefits of this in detail.
4. Abbreviations should be used in their full form the first time they appear in a paper, e.g. WQ and WR on line 87.
5. The units of the value range on line 397 should be appended.
6. Do not repeat references in the same sentence, such as line 95.
Reviewer 2 Report
Dear Authors,
your study is nicely done, and I have only minor comments, essentially at the beginning of the paper, where only an "engineering" perspective is given. As a physicist, I ask you to clearly separate clock "systematic" drift and ""random" effects from the start; to clearly justify how to choose the "best" clock; and why a genetic algorithm performs better than a purely "based-on-physics" algorithm. I attach my hand-written notes. Please also give references as "modern" as possible, in addition to the credit of pioneering works.

Reviewer 3 Report
The manuscript develops a method for the autonomous generation of high-precision time scales for navigation constellations. In brief, the paper presents a navigation constellation time scale generation framework and designs an improved Kalman plus weights (KPW) time scale algorithm and time-frequency steer algorithm that integrates genetic algorithms. The architecture of the navigation constellation high-precision time scale generation scheme is presented, followed by the improvement procedure of the existing time scale algorithm for obtaining a time scale with higher stability than a single clock. Eventually, the authors proposed an improved time-frequency steering algorithm to generate a comprehensive time scale that considers both short-term and long-term stability. The manuscript is well-written and worthy of publication in this journal. The work done is adequate and justifies the objectives considered in this work. From my perspective, it can be accepted for publication, but the only thing missing in the present manuscript is a comprehensive discussion section at the end of the results. Moreover, the authors may take another attempt to highlight the important extracts from this study in the conclusion section of the manuscript. The authors also may revisit the literature review for covering the progresses of similar works for better visibility of their manuscript. I have listed a few minor corrections below for implementation of the same in the subsequent version of the manuscript.
Minor comments:
Line 14 Incomplete sentence “ In particular, in the satellite autonomous navigation scenario,”
Line 21 KPW meaning should be expanded at its first appearance.
Line 27 Please verify the number in “ Allan deviation is 4,93E-16”
Line 57 I suggest the authors refer and cite Ansari and Jamjareegulgarn, 2022 (https://doi.org/10.1109/LGRS.2022.3204323) as the weighted average along with linear KF has a potential for such application.
Line 59 The authors may summarize in brief about capability of machine learning techniques. I realized the kernel extreme learning machine (KELM) could also be utilized like any other machine learning and deep learning approaches for improving the stability and performance in multi-constellation systems scenarios. The authors may have a review of https://doi.org/10.1038/s41598-023-27691-4 and https://doi.org/10.1007/s10509-022-04062-5 with appropriate citation and reference in the manuscript.
Line 102 The space in “In the second section ,” should be after the comma rather than prior to its position.
Line 198 The subsection heading should be corrected as “Genetic Algorithm”
Lines 367-371 Rewrite the sentence as in the present form it is grammatically meaningless.
Line 383 Please check the missing symbol within [ ].
Line 403, Line 517 and Line 529 Capitalize the first letter of the sentence in captions.
Line 533 A comprehensive Discussion section is missing before proceeding to the conclusion.
Line 542 Extra space in between “4. 93E-16” should be omitted.
